# The *RopGEF* Gene Family and Their Potential Roles in Responses to Abiotic Stress in *Brassica rapa*

**DOI:** 10.3390/ijms25063541

**Published:** 2024-03-21

**Authors:** Meiqi Zhang, Xiaoyu Wu, Luhan Chen, Lin Yang, Xiaoshuang Cui, Yunyun Cao

**Affiliations:** College of Horticulture Science and Engineering, Shandong Agricultural University, Tai’an 271018, China

**Keywords:** *B. rapa*, *RopGEF* genes, gene family, bioinformatics, expression profile, salt stress, osmotic stress

## Abstract

*Guanine nucleotide-exchange factors* (*GEFs*) genes play key roles in plant root and pollen tube growth, phytohormone responses, and abiotic stress responses. *RopGEF* genes in *Brassica rapa* have not yet been explored. Here, *GEF* genes were found to be distributed across eight chromosomes in *B. rapa* and were classified into three subfamilies. Promoter sequence analysis of *BrRopGEFs* revealed the presence of *cis*-elements characteristic of *BrRopGEF* promoters, and these *cis*-elements play a role in regulating abiotic stress tolerance and stress-related hormone responses. Organ-specific expression profiling demonstrated that *BrRopGEFs* were ubiquitously expressed in all organs, especially the roots, suggesting that they play a role in diverse biological processes. Gene expression analysis revealed that the expression of *BrRopGEF13* was significantly up-regulated under osmotic stress and salt stress. RT-qPCR analysis revealed that the expression of *BrRopGEF13* was significantly down-regulated under various types of abiotic stress. Protein–protein interaction (PPI) network analysis revealed interactions between RopGEF11, the homolog of *BrRopGEF9*, and the VPS34 protein in *Arabidopsis thaliana*, as well as interactions between *AtRopGEF1*, the homolog of *BrRopGEF13* in Arabidopsis, and the ABI1, HAB1, PP2CA, and CPK4 proteins. VPS34, ABI1, HAB1, PP2CA, and CPK4 have previously been shown to confer resistance to unfavorable environments. Overall, our findings suggest that *BrRopGEF9* and *BrRopGEF13* play significant roles in regulating abiotic stress tolerance. These findings will aid future studies aimed at clarifying the functional characteristics of *BrRopGEFs*.

## 1. Introduction

Guanine nucleotide-exchange factors (GEFs) activate ROP by converting GDP-bound ROP/RAC GTPases into their active GTP-bound forms, allowing them to interact with downstream effectors. Plant-specific RopGEF family members have a conserved RopGEF catalytic domain, the PRONE domain (plant-specific ROP nucleotide exchanger), and show catalytic activity towards multiple small G-protein ROPs [1]. These RopGEFs can dimerize and bind to two ROP molecules, and their activity is regulated by phosphorylation [2,3]. The catalytic activity of the PRONE structural domain can be inhibited by the C-terminus of RopGEFs. Recent research has shown that RopGEFs interact with RLKs on the membrane to relieve their inhibition of the PRONE domain, and the activated PRONE domain subsequently activates ROPs, which in turn regulates the expression of downstream genes [4,5].

Several studies have explored the role of *RopGEFs* in the abiotic stress responses of *A. thaliana*. The *A. thaliana* genome contains 14 *AtRopGEFs*, which exhibit a high degree of sequence similarity. The expression levels of these 14 *AtRopGEFs* are significantly altered in response to various abiotic stresses, such as cold, heat, salt, and osmotic stress. The expression of *AtRopGEF5* is up-regulated under salt and osmotic stress treatments but down-regulated under heat treatment; the expression of *AtRopGEF14* is increased under salt stress but decreased under heat stress. Furthermore, *AtRopGEF1*, *7*, *9*, and *12* levels are enhanced under heat stress but do not change in response to cold stress. However, osmotic stress induces the expression group 3 *AtRopGEFs*, such as *AtRopGEF1*, *5*, and *7*. Overall, these findings indicate that the expression of individual *AtRopGEFs*, as well as groups of *AtRopGEF* genes, is altered in various ways following exposure to abiotic stress [6].

Although few studies have been conducted on *RopGEFs* in other plant species, previous research on *A. thaliana*, *Oryza sativa*, *Medicago truncatula*, *Solanum lycopersicum*, and *Physcomitrium patens* indicates that they play a role in various developmental processes in plants [1,7]. In *A. thaliana*, *AtRopGEFs* play a role in various signaling-mediated developmental processes such as root development, stomatal development, and pollen tube growth. Specifically, *AtRopGEF1*, which is expressed at the tip of the pollen tube, plays a key role in maintaining normal polar growth by activating ROP1 during pollen tube elongation [7]. *AtRopGEF12* is also involved in the regulation of pollen tube polar growth [5]. The Pr form of the photosensitizing pigment activates the expression of *AtRopGEF11* (also known as *PIRF1*) in vitro under dark conditions, whereas the Pfr form of the photosensitizing pigment inhibits the expression of this gene. AtRopGEF11 can interact with ROPs in the presence of photosensitizing pigments to activate them, and these interactions play a key role in the regulation of light signals and the maintenance of normal root development [8]. *AtRopGEF1*, *AtRopGEF4*, and *AtRopGEF10* are involved in regulating stomatal development by affecting the activity of ROP11 in the ABA signaling pathway. Mutations of *AtRopGEF1* and *AtRopGEF4* result in increased sensitivity to ABA and increased stomatal closure [9]. Moreover, simultaneous mutations of *AtRopGEF1*, *AtRopGEF4*, and *AtRopGEF10* result in increased sensitivity to ABA. AtRopGEF7 regulates root tip stem cell homeostasis by regulating the expression of the growth hormone-mediated transcription factor PLT and affecting the accumulation of the growth hormone export protein PIN1, which affects the formation of growth hormone concentration gradients as well as the growth hormone response [10]. Additionally, *AtRopGEF1* controls the polar localization of the AUX1 protein and the accumulation and distribution of the PIN protein, which regulates embryonic development and the root gravitropic response [11]. A study of the upstream factors of *AtRopGEFs* revealed that plant-like receptor protein kinases RLKs (receptor-like kinases) are involved in *AtRopGEFs*-ROPs signaling and regulate downstream signaling by activating corresponding AtRopGEFs, which, in turn, activate ROPs [12]. The reduction in cell tip development is a significant macroscopic symptom of ARK mutants, and the growth phenotype is partially recovered by RopGEF3 expression and forced apical localization [13].

In rice (*O. sativa*), the roles of *OsRopGEFs* in several developmental processes, such as pollen germination, pollen tube growth, root development, and the expression of agronomic traits, such as plant height and grain length, have been clarified. Specifically, *OsRopGEF2*, *OsRopGEF3*, *OsRopGEF6*, and *OsRopGEF8* are predominantly expressed in pollen, indicating that they play key roles in pollen germination and pollen tube growth. Phenotypic analysis has shown that the double-knockout mutants *OsRopGEF2* and *OsRopGEF8* are characterized by a significant reduction in pollen germination and seed yield [14]. OsRopGEF3 was found to interact with OsRac3 to regulate root hair elongation and reactive oxygen species (ROS) production. OsRopGEF3 converts the GDP of OsRac3 to GTP, and the activated OsRac3 protein interacts with OsRBOH5 to generate ROS, indicating that OsRopGEF3 plays a key role in rice root hair growth [15]. Furthermore, single mutants of *OsRopGEF5* and *OsRopGEF3* have been shown to affect plant height and grain length, respectively [16]. A previous study of *M. truncatula* has revealed that the down-regulated expression of *MtRopGEF2* results in shortened root hairs similar to those observed in DN-MtROP3. This finding suggests that MtRopGEF2 plays a role in root hair growth by affecting the activity of MtROPs [17]. In tomato (*S. lycopersicum*), the kinase partner protein (KPP) serves as an ROP guanine nucleotide exchange factor (GEF) responsible for activating ROP GTPases and interacts with the tomato pollen receptor kinases LePRK1 and LePRK2. KPP nucleates branched actin by recruiting it via the membrane-localized receptors LePRK1 and LePRK2, the ARP2/3 complex, and actin bundles, and this plays a key role in regulating pollen tube growth and shape [4]. In *P. patens*, RopGEFs and ROP GTPASE-ACTIVATING PROTEINs (RepGAPs) form membrane domains when they grow in the tips of cells [18].

Chinese cabbage (*Brassica rapa*), an economically important crop with high nutritional value, is highly susceptible to the deleterious effects of abiotic stresses, especially salt and osmotic stress. Diverse molecular mechanisms underlie the deleterious effects of these stresses on plants [19,20]. However, the roles of AtRopGEFs in development and stress responses in *B. rapa* have not yet been elucidated. Thus, *BrRopGEF* gene family members were identified and annotated using bioinformatic approaches, and their structures, protein sequences, conserved structural domains, evolutionary relationships, and expression patterns were systematically analyzed to aid future studies of the biological functions of *BrRopGEF* gene family members in *B. rapa*.

## 2. Results

### 2.1. Identification of RopGEF Genes and Analysis of the Physical–Chemical Properties and Subcellular Localization of Their Encoded Proteins

We used HMMER and local BLAST algorithms to compare *BrROPGEF* sequences against the *B. rapa* genome database. We confirmed the presence of the PRONE (or PRONE superfamily) domains using the conserved domain database (CDD) and protein families (PFAM) database. A total of 21 *BrRopGEFs* were identified. Based on their physical positions in the *B. rapa* genome, these *BrRopGEF* members were named *BrRopGEF1*–*BrRopGEF21* (Table 1). Among the 21 *BrRopGEF* genes, the length of *BrRopGEF1* was the shortest (1586 base pairs), and the length of *BrRopGEF17* was the longest (7369 base pairs). The lengths of the amino acid sequences ranged from 391 to 941 amino acids, and the corresponding molecular masses of the proteins ranged from 44,107.32 to 565,501.00 Daltons. The isoelectric points of the BrRopGEF proteins in *B. rapa* varied from 4.95 to 9.12. The isoelectric points of 18 BrROPGEF proteins were greater than 7, indicating that the number of acidic amino acids was higher in BrROPGEF proteins than in other proteins within the family. Subcellular localization predictions, which were performed using the WOLF PSORT server, indicated that BrRopGEFs were distributed across various organelles within the cell, including the nucleus, cytosol, and chloroplast.

### 2.2. Chromosomal Mapping Analysis of BrRopGEF Genes in B. rapa

Chromosome mapping analysis of *BrRopGEF* genes revealed that a total of 21 *BrRopGEFs* were distributed across eight chromosomes (Figure 1). The greatest number of *BrRopGEF* genes (5) was observed on Chromosome 5. Four *BrRopFEG* genes were observed on Chromosome 9. Chromosomes 3, 7, 9, and 10 each had three *BrRopGEF* genes, and Chromosomes 3 and 6 each contained two *BrRopGEF* genes. Chromosomes 1, 2, and 6 each contained a single *BrRopGEF* gene. *BrRopGEF7* and *BrRopGEF8* were located next to each other on Chromosome 5, suggesting that these genes are tandemly duplicated pairs.

One of the aims of our study was to elucidate the phylogenetic relationships among *BrRopGEF* genes; we constructed a phylogenetic tree using genes from *B. rapa*, *A. thaliana*, and *O. sativa*. *B. rapa* was more closely related to *A. thaliana* than to *O. sativa* (Figure 2A). Based on the phylogenetic tree, we categorized the 21 *BrRopGEFs* into six distinct groups (I–VI). Group I comprised six *B. rapa* genes, including two from *A. thaliana*. Group II comprised eight genes, including five from *B. rapa* and three from *A. thaliana*. Group III comprised eight members, three from *B. rapa*, three from *O. sativa*, and one from *A. thaliana*. Group IV was the largest group, comprising eleven members, including four from *B. rapa*, three from *O. sativa*, and four from *A. thaliana*. Therefore, the phylogenetic analysis suggested that *RopGEFs* in *B. rapa* were more closely related to those in *A. thaliana* than to those in *O. sativa*.

We examined collinearity relationships between *BrRopGEF* genes and genes of *A. thaliana* and *O. sativa* (Figure 2B). Strong collinearity relationships were observed between 18 *BrRopGEFs* and 14 genes in *A. thaliana* and between 3 *BrRopGEF* genes and 2 genes in *O. sativa*.

### 2.3. Analysis of Promoter Cis-Regulatory Elements of RopGEFs in B. rapa

We conducted a comprehensive analysis to elucidate the potential mechanisms underlying the regulation of *BrRopGEF* genes, as well as the influence of phytohormones and stress-responsive elements on their regulation. The PlantCARE web server was used to identify putative cis-elements specific to the promoters of *BrRopGEF* genes. A total of 18 prominent regulatory *cis*-acting regulatory elements (CAREs) were identified. These elements have been shown to play a role in various physiological processes, including auxin responses, abscisic acid (ABA) responses, gibberellin (GA) responses, methyl-jasmonate (MeJA) responses, low-temperature responses, and drought responses (Figure 3). The promoters of all *BrRopGEF* genes, with the exception of the promoters of *BrRopGEF11* and *BrRopGEF13*, contained anaerobic-inducible regulatory elements. Further examination revealed that approximately 67% of the CAREs contained ABA elements, 62% contained MeJA-response elements, around 52% contained GA-responsive elements, approximately 43% contained drought-induced response elements, and 33% contained low-temperature-responsive elements.

### 2.4. Analysis of BrRopGEF Gene Structure and the Domain Distribution of Their Encoded Proteins

A comprehensive analysis was conducted to investigate the structural characteristics of *BrRopGEFs* through examination of their exon architectures (Figure 4A). Substantial variation in exon architecture was observed among *BrRopGEF* genes, and the number of exons ranged from 4 to 16, indicating that considerable divergence was observed within the *BrRopGEF* family.

Phylogenetic analysis revealed that *BrRopGEFs* can be classified into two main groups with distinct conserved motifs (Figure 4A). Class 1 contained three *BrRopGEF* genes; *BrRopGEF7* contained Motif 1, Motif 2, Motif 5, and Motif 7; *BrRopGEF8* contained Motif 1 to Motif 7; and *BrRopGEF17* contained Motif 1 to Motif 8. Category 2 contained 18 genes, and these could be further subdivided into two distinct groups: Group 1, which contained seven *BrRopGEF* genes containing Motif 1 to Motif 9, and Group 2, which contained 11 *BrRopGEF* genes, all of which lacked Motif 1, with the exception of *BrRopGEF1*. The remaining 10 genes in Group 2 contained 10 motifs (Motif 1 to Motif 10).

The number of conserved motifs within each *BrRopGEF* gene ranged from 4 to 10. Most *BrRopGEF* genes had 7 to 10 motifs; however, *BrRopGEF7* had only 4 conserved motifs. All *BrRopGEF* genes had Motif 1, with the exception of *BrRopGEF1*; thus, 95.2% of the *BrRopGEF* genes contained Motif 1. Additionally, all *BrRopGEF* genes contained Motif 2, Motif 5, and Motif 7. These findings strongly suggest that Motif 1, Motif 2, Motif 5, and Motif 7 are the most conserved motifs within the *BrRopGEF* gene family (Figure 4A). Consequently, the arrangement of structural motifs varies among members of the *BrRopGEF* family, but is highly similar among closely related genes.

*BrRopGEF* genes encode proteins with two primary conserved domains, including PRONE domains (or the PRONE superfamily). The presence of PRONE domains is essential for AtRopGEF to catalyze the conversion of GDP to GTP (Figure 4B) [9]. Similar protein structures were observed within the *BrRopGEF* family, as evidenced by their placement in the same phylogenetic branch, which suggests that they have shared functions. Examination of motif logos and information regarding the motifs of *BrRopGEFs* (Figure 4C and Table 2) revealed a prevalence of conserved amino acids within all motifs, which underscores their importance for protein function.

### 2.5. Gene Ontology (GO) Annotation Analysis of BrRopGEFs

We conducted a GO analysis of *BrRopGEF* genes to elucidate their potential functions. *BrRopGEF* genes were enriched in 39 GO terms in three categories: “Biological Processes, GO-BPs”, “Cell Components, GO-CCs”, and “Molecular Functions, GO-MFs” (Figure 5, Appendix A). The most enriched GO terms were guanyl-nucleotide exchange factors (GO:0005085), GTPase regulators (GO:0030695), nucleoside-triphosphatase regulator activity (GO:0060589), and enzyme regulator activity (GO:0030234). In the cellular component (CC) category, the most enriched GO terms were plasma membrane (GO:0005886), apical part of the cell (GO:0045177), and cell periphery (GO:0071944), and this was consistent with subcellular localization prediction. The main GO terms in the biological process category were cellular component organization (GO:0016043), cellular component organization or biogenesis (GO:0071840), and other related processes. Overall, the GO data indicated that *BrRopGEFs* play a key role in the regulation of gene expression.

### 2.6. Gene Expression Analysis of BrRopGEFs

#### 2.6.1. Analysis of the Organ-Specific Expression of BrRopGEF Genes in Organs

Expression patterns of the 21 *BrRopGEFs* in five organs of *B. rapa* (root, stem, leaf, flower, and callus) were investigated (Figure 6, Appendix A). The similarity of the expression patterns among genes was positively associated with their sequence similarity. *BrRopGEF1*, *BrRopGEF15*, *BrRopGEF19*, and *BrRopGEF20* were highly expressed in roots. Only *BrRopGEF13* and *BrRopGEF21* were significantly expressed in stems, suggesting that they play a role in stem growth and development processes. Moreover, several *BrRopGEF* genes were highly expressed in flowers, suggesting that they play a role in regulating flowering-related traits, such as flowering time and duration. Overall, these findings highlight the diverse roles of different *BrRopGEFs* in the developmental processes of various organs.

#### 2.6.2. Analysis of BrRopGEFs Involved in Responses to Abiotic Stress

The growth and yield of plants and crops are hampered by the osmotic stress caused by drought and high salinity [21]. We used transcriptome analysis, proteomic assays, and RT-qPCR analysis to determine changes in the expression of *BrRopGEFs* under osmotic stress and salt stress.

Analysis of transcriptomic data from drought-sensitive (DS) and drought-tolerant (DT) plants revealed differential expression patterns, especially for *BrRopGEF13*, *17*, *20*, and *21* (Figure 7A, Appendix A). The expression levels of *BrRopGEF13*, *BrRopGEF20*, and *BrRopGEF21* were lower in DT plants than in DS plants. Both *BrRopGEF13* and *BrRopGEF20* contain GA-response elements, and *BrRopGEF20* and *BrRopGEF21* also contain ABA-response elements. These findings support the hypothesis that the up-regulation of these genes in response to osmotic stress is associated with GA and ABA metabolism [22].

We investigated the expression levels of *BrRopGEFs* under control conditions (0 h) and osmotic stress (6 h) using RNA-seq analysis (Figure 7B, Appendix A). Our findings revealed that the expression of nine genes was up-regulated, and the expression of two genes was down-regulated. The expression of the other genes was not significantly altered under osmotic stress compared with control (CK) conditions. We analyzed the expression of several genes using quantitative real-time PCR (RT-qPCR) based on their expression patterns in roots and stems under osmotic stress, which were inferred from RNA-seq analysis. The expression of *BrRopGEFs* was generally up-regulated at 2–4 h, and no significant differences in the expression of *BrRopGEFs* were observed between treatments and the control at 12 h (Figure 7C, Appendix A). The expression of *BrRopGEF17* was substantially up-regulated 10–20-fold at 4–6 h compared with the control. Conversely, the expression of *BrRopGEF13* and *21* decreased after being initially up-regulated, and their expression at 12 h under osmotic stress was lower than that under control conditions; this was consistent with the results of the RNA-seq analyses.

We found that the expression of nine genes was down-regulated under salt stress conditions. In contrast, the expression of *BrRopGEF4* was up-regulated under salt stress conditions (Figure 8A, Appendix A).

We also performed RT-qPCR analysis (Figure 8B, Appendix A). The results revealed that the expression of *BrRopGEF4* and *19* was significantly up-regulated under salt stress compared with the CK. The expression of *BrRopGEF13* and *BrRopGEF17* was significantly down-regulated under salt stress. These RT-qPCR findings are consistent with the transcriptome data, confirming their reliability.

### 2.7. Analysis of the Protein Secondary and Tertiary Structures of BrRopGEFs

The protein secondary structures of BrRopGEFs in *B. rapa*, including α-helixes, extended chains, β-turns, and random coil components, were analyzed. The most common secondary structures in the RopGEF proteins were α-helixes and random coil components; β-turns were the least common components (Figure 9A).

Furthermore, the tertiary structures of proteins are determined by additional coiling and folding processes based on the secondary structures. Visualization of the tertiary conformations can provide insights into the structural characteristics of these proteins and their evolutionary relationships. We used computational techniques to predict the three-dimensional protein structure of RopGEF1, using proteins encoded by homologs of *BrRopGEF* in *A. thaliana* as a representative model. Two different angles of the resulting model are shown, and the distinct colors denote the various secondary structures and labeling of specific structural domains (Figure 9B). Additionally, homology modeling was used to predict the tertiary structures of ROPGEFs in *B. rapa* (Figure 9C), and this revealed a notable degree of structural similarity among members of the same subgroup. This observation indicates that homologous structures have been evolutionarily maintained.

The PRONE domain is divided into three subdomains that are highly conserved and separated by two short segments of variable amino acid residues [1,7]. The molecule is approximately 370 residues in length and mainly consists of alpha-helices, with the exception of a beta-turn that protrudes from the main body of the molecule. The PRONE domain’s structure is divided into two subdomains: SD1, which includes the helices alpha1-5 and alpha13, and SD2, which includes the helices alpha6-12 [3]. The interaction between the PRONE domain and the nucleotide binding site of the RopGEF protein leads to the release and exchange of nucleotides [23].

### 2.8. Protein–Protein Interaction (PPI) Network Analysis of BrRopGEFs

Proteins play a key role in carrying out cellular and tissue functions, and they are involved in diverse life activities [24]. PPI networks consist of proteins that interact with each other and participate in various life processes, such as biological signaling, gene expression regulation, energy and material metabolism, and cell cycle regulation [24,25,26,27]. Identifying the links between unknown functional proteins and PPI interaction networks can provide insights into the complex biological functions of proteins and the dynamic regulation of network interactions among biomolecules within cells [28,29]. We constructed predicted PPI network maps for ROPGEFs using the integrated resources and algorithms available in the STRING database. Strong interactions of ROPGEF members in *A. thaliana* with RAC10, PRK2A, ROP1, RAC2, and ARAC5 were detected (Figure 10A, Appendix A). *BrRopGEF13* was orthologous to *AtROPGEF1*, and *BrROPGEF9* and *BrROPGEF11* were orthologous to *AtROPGEF1*. In *A. thaliana*, *AtRopGEF1* specifically regulates the function of ROP11 in ABA-mediated stomatal closure [30].

Analysis of the PPI network revealed an intriguing similarity between the predicted PPI networks of RopGEF13 in *A. thaliana* and RopGEF1, which suggests that transcription factors involved in ABA signaling, including ABSCYACID INSENSITIVE1 (ABI1), HYPERSENSITIVE TO ABA1 (HAB1), and protein phosphatase 2CA (PP2CA), interact with the proteins encoded by homologs of *BrRopGEF13* (Figure 10B). Furthermore, the protein encoded by the homolog of *BrRopGEF9* and *BrRopGEF11* (*AtRopGEF11*) was found to interact with phosphatidylinositol 3-kinase (*AtVSP34*) in *A. thaliana* (Figure 10C). These findings suggest that the proteins encoded by *BrRopGEFs* engage in diverse interactions with proteins encoded by different gene families to mediate responses to abiotic stresses.

## 3. Discussion

RopGEFs are a highly conserved protein family specific to plants, and they play key roles in mediating the adaptation of plants to abiotic stress. In this study, we aimed to identify *RopGEFs* and analyze their structure and expression patterns, as well as the sequences of their encoded proteins, conserved structural domains, and evolutionary relationships. The aim was to elucidate the role of *BrRopGEFs* in the responses of *B. rapa* to abiotic stress.

*BrRopGEF13* potentially plays a role in regulating the response of *B. rapa* to osmotic stress. Comprehensive analysis of its promoter elements revealed GA-responsive motifs within the *BrRopGEF13* promoter region. Previous studies have revealed that osmotic conditions result in the deactivation of GA, which induces premature stomatal closure in response to soil desiccation. This, in turn, inhibits the synthesis of GA within leaf tissues, thereby restricting the transpirational surface area [31]. ABA plays a role in the abscission of plant leaves and responses to osmotic stress [32]. These findings indicate that GA and ABA play a role in adaptation to osmotic stress. The down-regulation of these genes was observed in both the RNA-seq and RT-qPCR analyses.

According to the PPI network analysis, the homolog of *BrRopGEF13*, *AtRopGEF1*, interacted with protein phosphatases, such as ABI1, HAB1, and PP2CA, as well as calcium-dependent protein kinases (CPKs) [33]. Active CPKs phosphorylate GEF1, which mediates the transport and degradation of GEF1. The removal of GEF1 leads to the deactivation of ROP10 and ROP11, which serve as negative regulators in the ABA signal transduction pathway [34,35,36]. Hence, the removal of ABA-induced GEF by CPKs promotes ABA signal transduction by dismantling the negative regulatory loop composed of PP2C-GEF-ROP10/ROP11 [30,34,35,36]. In conclusion, the *BrRopGEF* gene might play a role in the response to osmotic stress.

*BrRopGEF9* likely plays a key role in the response of *B. rapa* to osmotic stress and salt stress. Analysis of *cis*-acting elements within the promoter region of *BrRopGEF9* reveals the presence of SA and ABA-responsive elements. Under osmotic conditions, ABA, which acts as a chemical signal, induces stomatal closure, thereby promoting the responses of plants to osmotic stress and salt stress [37]. The biosynthesis of SA can also enhance the tolerance of plants to osmotic conditions [38]. The transcriptome data and RT-qPCR results consistently indicated that the expression of *BrRopGEF9* is up-regulated in response to changes in salt levels.

In *A. thaliana*, the homologous gene *AtRopGEF11* has been demonstrated to interact with VSP34 (phosphatidylinositol 3-kinase). GO analysis suggests that VSP34 is involved in processes related to stomatal closure, ROS metabolism, and salt stress response regulation. *AtRopGEF11* is a component of the COP9 Signalosome (CSN), which is associated with various cellular and developmental processes, including photomorphogenesis, auxin, JA responses, and PPIs [39]. Previous studies have indicated that JA effectively mitigates salt-stress-induced damage in plants [40,41]. RopGEF11 can interact with ROPs and induce their activation, which regulates light signaling and maintains normal root development [8]. Based on these findings, we suggest that *BrRopGEF9* might play a key role in the response of *B. rapa* to osmotic stress and salt stress through its effects on root development, osmotic response elements, hormone regulation, and protein interactions.

## 4. Materials and Methods

### 4.1. Identification of RopGEF Genes and Analysis of the Physical-Chemical Properties and Subcellular Localization of Their Encoded Proteins

*A. thaliana* genomic data were acquired from TAIR (https://www.arabidopsis.org/ (accessed on 14 December 2022)), and *B. rapa* genomic data were downloaded from BRAD (http://brassicadb.cn/ (accessed on 23 December 2022)) to construct a local BLAST database [42]. Through bidirectional BLAST searches (E-value < 1 × 10^−10^, Identity > 40%), 21 known *BrRopGEFs* were identified. A conserved domain database from NCBI was used to analyze the conserved domains of the selected members. (https://www.ncbi.nlm.nih.gov/cdd (accessed on 24 December 2022)) [43], and the domains were verified using Pfam (E-value < 1.0) (http://pfam.xfam.org/ (accessed on 24 December 2022)) [44]. Members that did not contain typical domains for *BrRopGEFs* were eliminated.

The Bioinformatics Resource Portal ExPASy server (http://web.expasy.org/protparam/ (accessed on 3 January 2023)) with default parameters was used to determine biochemical properties, such as amino acid composition, molecular weight (MW), and theoretical pI, for BrRopGEF proteins [45]. The SWISS-MODEL (https://swissmodel.expasy.org/ (accessed on 15 January 2023)) online program was used to predict the structure of BrRopGEF proteins using default parameters [46].

The WOLF PSORT website (https://www.genscript.com/wolf-psort.html (accessed on 21 January 2023)) was used to conduct subcellular localization analysis [47].

### 4.2. Chromosome Localization, Multiple Sequence Alignment, and Phylogenetic Analysis

According to the *B. rapa* genome information available on BRAD (http://www.brassicadb.cn/ (accessed on 14 February 2023)), the chromosomal locations and duplications of *BrRopGEF* genes were mapped, and the gene density information of each chromosome was obtained using TBtools (v1.120). The chromosomal distribution of genes and gene duplication events were analyzed using the Multiple Collinearity Scan toolkit (One Step MCscanX-Super Fast) in TBtools (v1.120) [42].

*AtRopGEFs* were retrieved from The Arabidopsis Information Resource (TAIR) database (https://www.arabidopsis.org/ (accessed on 28 February 2023)). Multiple sequence alignment of *BrRopGEFs* and *AtRopGEFs* was performed using Clustal X version 2.1. Subsequently, phylogenetic trees were constructed using the maximum likelihood (ML) method in MEGA 11.0 [48]. ML analyses were performed, and statistical support for each node was estimated using bootstrap analysis with 1000 replicates. The phylogenetic tree was generated using Interactive Tree of Life (IToL) v. 4 (https://itol.embl.de/ (accessed on 11 March 2023)), and scale bars represent substitutions of 0.1 [49]. The proteins encoded by *AtRopGEFs* and *OsRopGEFs* were used as outgroups. We downloaded protein sequences from the TAIR (https://www.arabidopsis.org/ (accessed on 18 March 2023)) and Phytozome 13 (https://phytozome-next.jgi.doe.gov/ (accessed on 18 March 2023)) databases for Arabidopsis and rice, respectively [50].

### 4.3. Analysis of CAREs

To identify CAREs, 2000 bp upstream sequences of *BrRopGEFs* were downloaded from Ensemble Plants and analyzed using PlantCARE (http://bioinformatics.psb.ugent.be/webtools/plantcare/html (accessed on 26 March 2023)) [51]. The most common CAREs were identified using TBtools (v1.120). After determining the number of *cis*-acting elements on each promoter, the expected *cis*-acting elements were classified according to their function, and tables were created using Excel 2021.

### 4.4. Motif Analysis and GO

The MEME program (http://meme-suite.org/tools/meme (accessed on 8 April 2023)) was used for the identification of conserved motifs using the following parameters: number of repetitions, any; maximum number of motifs, 10. The results were displayed using TBtools (v1.120) [52].

The GO is an internationally standardized ontology of terms for describing the functions of genes and gene products in organisms. Our functional enrichment tests of the candidate genes were conducted using the GO analysis online server g:Profiler (https://biit.cs.ut.ee/gprofiler/gost (accessed on 19 April 2023)) [53].

### 4.5. Expression Profiling of BrRopGEF Genes

RNA-seq data with organ-specific expression (GEO: GSE43245) and gene expression profiles of drought-tolerant and sensitive *B. rapa* under control and drought stress conditions (GEO: GSE73963) were obtained via transcriptome sequencing of *B. rapa* (http://brassicadb.cn (accessed on 12 May 2023)). All transcriptome expression data were log_2_-transformed (Appendix A). Cluster heatmaps and bar graphs were made to visualize the differential expression of gene family members under control and osmotic stress conditions. The above were mapped using TBtools (v1.120) and Excel 2021 [54,55].

### 4.6. Total RNA Extraction and RT-qPCR

In the RNA-seq analysis, two-week-old, stable, self-incompatible Chinese cabbage seedlings cultivated in Hoagland nutrient solution (Coolaber, Beijing, China) were subjected to stress conditions. Specifically, they were treated with 15% PEG 6000 and 150 mM NaCl for 12 h to simulate osmotic stress and salt stress, respectively. Untreated two-week-old Chinese cabbage seedlings were used as control samples.

*B. rapa* with stable self-incompatibility was utilized for the stress treatments. The plump seeds were seeded in MS modified medium (with vitamins, sucrose, and agar) and cultivated in a plant incubator. Seedlings with six leaves and similar growth statuses were selected for the stress treatments. Subsequently, they were subjected to stress conditions through treatment with 15% PEG 6000 and 150 mM NaCl for 2, 4, 6, and 12 h to simulate osmotic stress and salt stress, respectively. A control group of Chinese cabbage seedlings was maintained under normal conditions. Each treatment comprised three biological replicates, and all samples were cryopreserved at −80 °C for subsequent RNA extraction.

Total RNA was extracted using a FastPure^®^ Cell/Tissue Total RNA Isolation Kit V2 (Vazyme Biotech Co., Ltd., Nanjing, China); RT-qPCR was performed using the TransScript Uni All-in-One First-Strand cDNA Synthesis SuperMix (TransGen. AU341-02, Beijing, China). RT-qPCR primer sequences were designed using the qPrimerDB-qPCR primer database (https://biodb.swu.edu.cn/qprimerdb/ (accessed on 31 May 2023)) [56]. RT-qPCR was performed on a qTOWER3 qPCR machine using TransStart^®^ Green qPCR SuperMix (TransGen Biotech, Beijing, China) and *BrActin2* as the reference gene; expression levels were determined using the 2^−ΔΔCt^ method. Reactions were performed with three technical replicates, and data were analyzed using Excel 2021. The specific primer sequences are listed in Appendix A.

### 4.7. Analysis of Protein Secondary Structure, Tertiary Structure, and PPI Networks

PRABI was used to analyze the secondary structure of BrRopGEF proteins (https://prabi.ibcp.fr/htm/site/web/app.php/home (accessed on 15 June 2023)), and the results were analyzed using Excel 2021.

The tertiary structure of BrRopGEF proteins was predicted and analyzed using SWISS-MODEL (https://swissmodel.expasy.org/ (accessed on 7 July 2023)), and tertiary structure images of each member were compiled using Adobe Photoshop CC 2023 [57].

For PPI network analysis, the PPI network prediction website (https://cn.string-db.org/ (accessed on 22 July 2023)) was used to obtain *B. rapa* PPI network maps (minimum interaction score = 0.150; default parameters were used for other parameters) [58].

To examine the BrRopGEF PPI network, we used the STRING online server with default parameters [58]. The PPI network was constructed using Cytoscape v3.9.1 [59].

## 5. Conclusions

In this study, a total of 21 *BrRopGEFs* were identified within the *B. rapa* genome. Sequence analysis, *cis*-element identification, expression profiling across various organs, assessment of abiotic stress tolerance, and a thorough review of previous studies indicated that *BrRopGEF9* could play a key role in regulating salt stress tolerance. Our findings also indicate that *BrRopGEF13* plays a key role in the response to osmotic stress and salt stress.

## Figures and Tables

**Figure 1 ijms-25-03541-f001:**
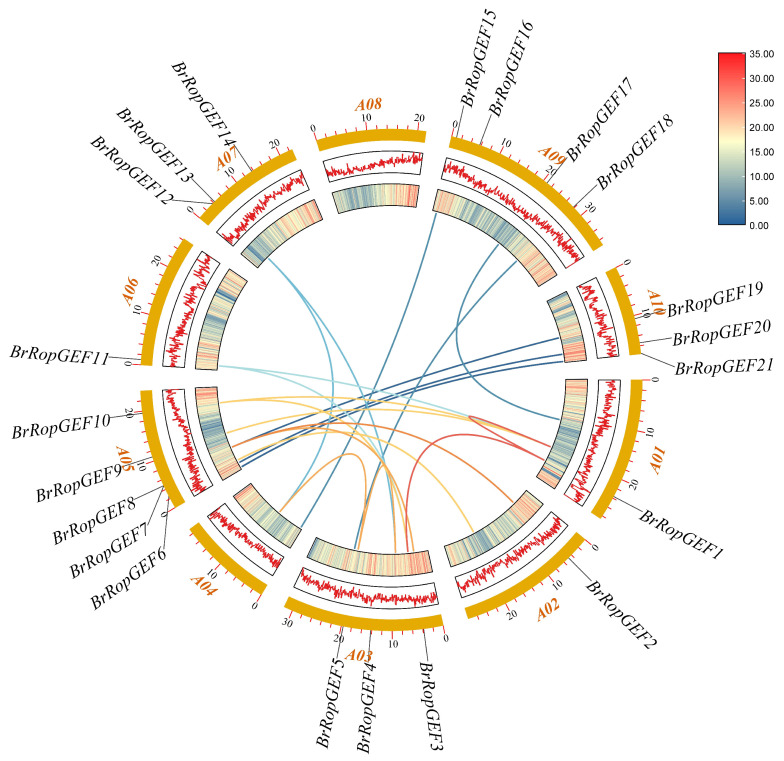
Localization and replication of *BrRopGEFs* on chromosomes in *B. rapa*. The outermost blocks, colored yellow, represent the 10 chromosomes of *B. rapa*. The middle and inner regions show the density of each chromosome in the line and heatmap formats. The colored lines in the middle indicate the collinearity relationships among *BrRopGEFs* within *B. rapa*.

**Figure 2 ijms-25-03541-f002:**
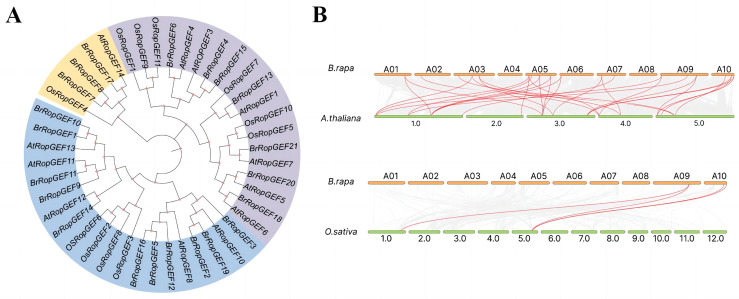
Phylogenetic relationships and synteny analysis of *RopGEF* genes in *B. rapa*, *A. thaliana*, and *O. sativa*. (**A**) Analysis of the phylogenetic relationships among *RopGEFs* of *B. rapa*, *A. thaliana*, and *O. sativa*. Group I is denoted by purple characters, and Group II is represented by blue characters. Yellow characters indicate Group III. (**B**) Analyses of gene expression patterns between *B. rapa* and other plants (*A. thaliana* and *O. sativa*). Red lines indicate homologous gene pairs.

**Figure 3 ijms-25-03541-f003:**
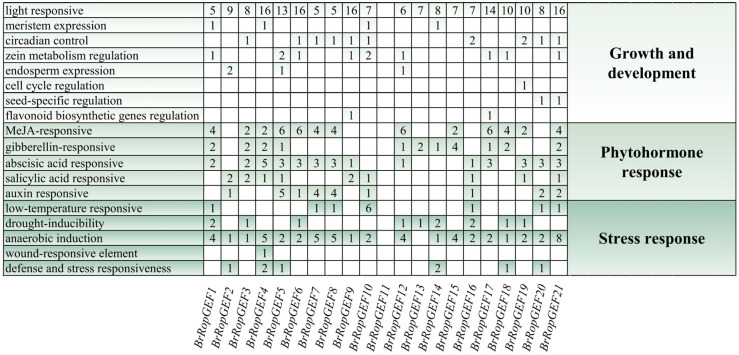
Analysis of *cis*-elements in promoters of *BrRopGEFs*. Different shades of green indicate the presence of *cis*-elements involved in different biological processes, and the number in each cell indicates the number of *cis*-acting elements in each gene. White cells indicate the absence of the *cis*-acting element. Different shades of green on both sides of the square correspond to distinct biological processes.

**Figure 4 ijms-25-03541-f004:**
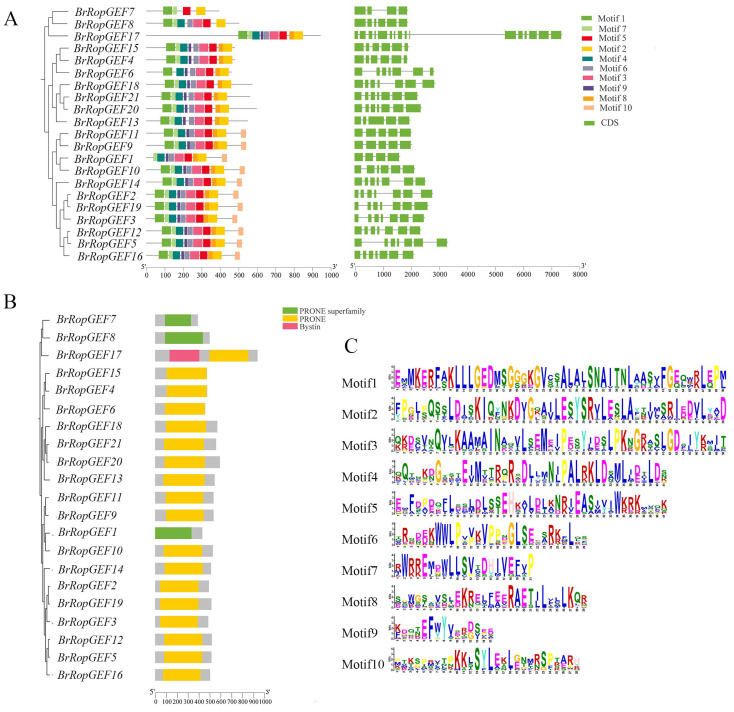
Motifs and gene structures. (**A**) Structures of *BrRopGEF* genes and conserved structural domains. Green boxes indicate exons, and a MEME analysis identified the conserved domain of RopGEF in *B. rapa*. (**B**) Each color indicates a specific domain. (**C**) The conserved motifs of RopGEFs in *B. rapa*. The integral height of each stack indicates the level of conservation at this site, and the amino acid frequency is indicated by the size of each letter.

**Figure 5 ijms-25-03541-f005:**
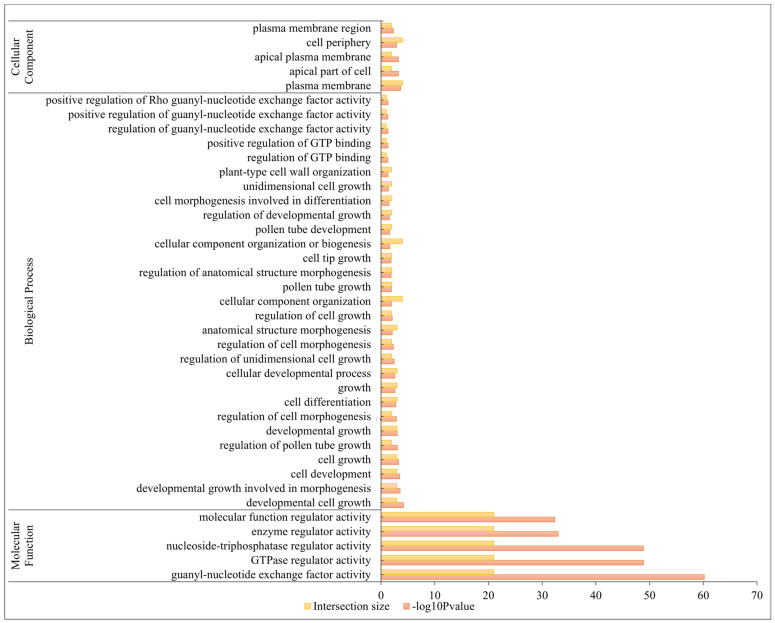
GO annotation of *BrRopGEFs*. The GO terms were derived from three categories: “BP” (biological process), “CC” (cellular component), and “MF” (molecular function). The x-axis denotes the −logPvalue, and the y-axis shows the corresponding GO terms. The size and color of the bars indicate the number of genes associated with each specific GO term.

**Figure 6 ijms-25-03541-f006:**
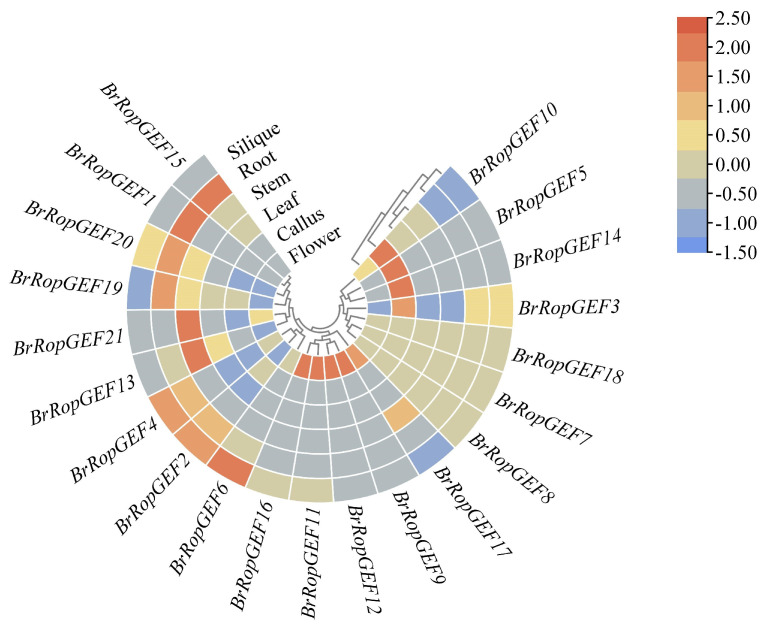
Heatmap showing the expression patterns of *BrRopGEFs* in different organs of *B. rapa.* Cells indicate the log_2_-transformed expression levels of genes in different organs, with darker shades of red indicating higher expression levels and bluer shades indicating lower expression levels.

**Figure 7 ijms-25-03541-f007:**
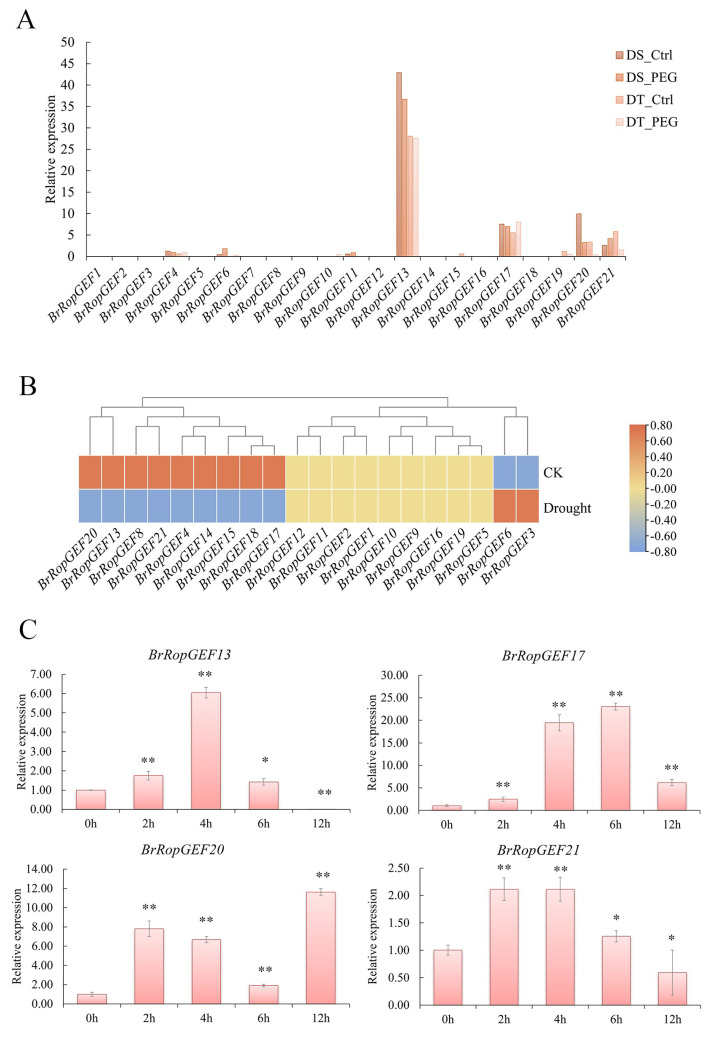
Transcriptome analysis reveals osmotic stress-induced changes in the expression of *BrRopGEFs.* (**A**) Analysis of changes in the expression of *BrRopGEFs* under osmotic stress. The PEG treatment for drought-tolerant and drought-sensitive plants is denoted by DT-PEG and DS-PEG, respectively. (**B**) Expression of *BrRopGEFs* in *B. rapa* under PEG-6000 treatment for 0 h and 6 h as determined by RNA-seq. (**C**) Expression of *BrRopGEFs* according to RT-qPCR analysis. Asterisks above the bars indicate the presence of significant differences (two-tailed *t*-test * *p* < 0.05, ** *p* < 0.01), respectively.

**Figure 8 ijms-25-03541-f008:**
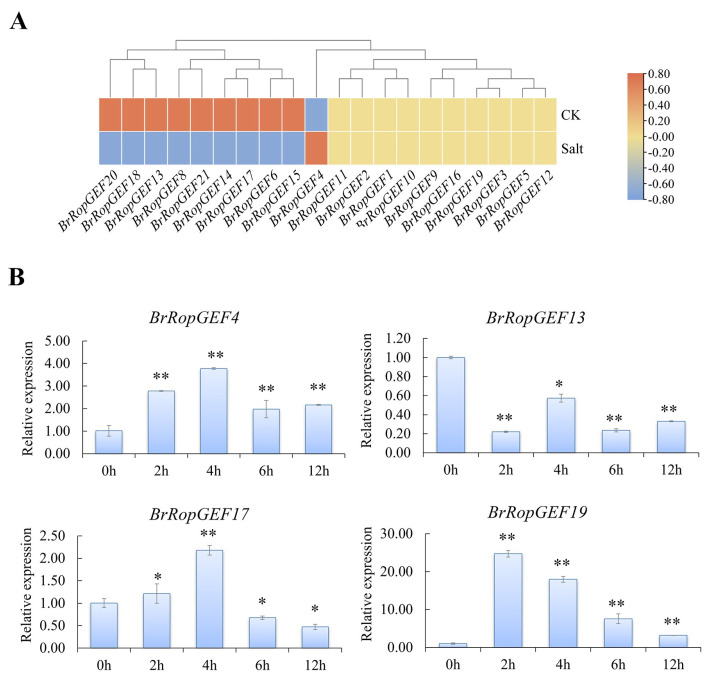
Analysis of the expression levels of *RopGEFs* in *B. rapa* under salt stress. (**A**) Heatmap of *BrRopGEFs* under salt stress. (**B**) Expression of *BrRopGEFs* inferred by RT-qPCR analysis. Asterisks above the bars indicate the presence of significant differences (two-tailed *t*-test * *p* < 0.05, ** *p* < 0.01), respectively.

**Figure 9 ijms-25-03541-f009:**
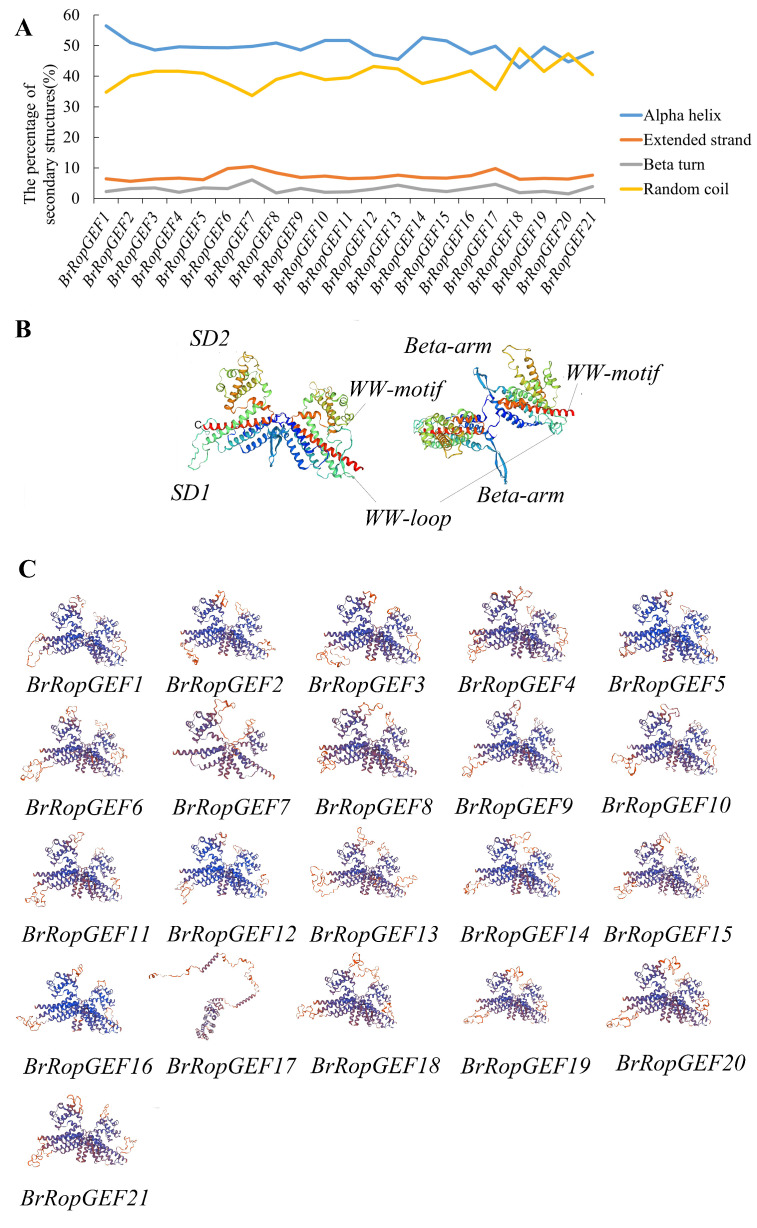
Analysis of the secondary and tertiary structures of ROPGEFs in *B. rapa*. (**A**) Secondary structure prediction. (**B**) Physicochemical predictions of the tertiary structure of ROPGEF1 in *A. thaliana*, which has a three-dimensional butterfly shape. Each protomer consisted of two subdomains: SD1 and SD2. Lines of different colors and shapes represent the various secondary structure of the protein in different places. (**C**) Prediction of tertiary structures. Different subunits are shown in different colors.

**Figure 10 ijms-25-03541-f010:**
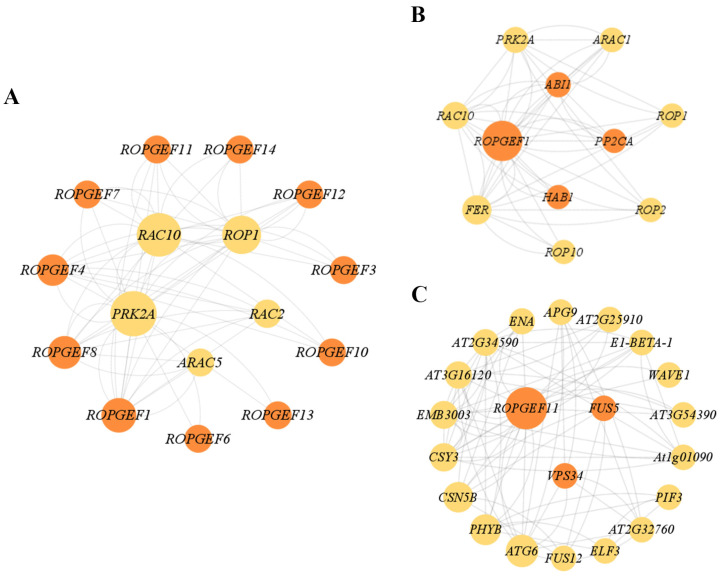
Predictive PPI interaction network for ROPGEFs in *A. thaliana*. (**A**) Predictive analysis of ROPGEF interaction networks in *A. thaliana*. (**B**) Predictive analysis of the ROPGEF1 in *A. thaliana*. (**C**) Predictive analysis of the interaction network of ROPGEF11 in *A. thaliana*. The minimum engagement score requirement is 0.150; default settings were used for other parameters. Network nodes represent proteins, and edges represent protein–protein associations. The circle size indicates the degree value of the node, and the larger the circle, the higher the degree value.

**Table 1 ijms-25-03541-t001:** Physicochemical properties of the proteins encoded by *BrRopGEF* genes.

Gene Name	Gene ID	Chromosome	pI	MW (Da)	Protein Length (aa)	Subcellular Location	*A. thaliana* ID	*A. thaliana* Name
*BrRopGEF1*	*Bra021162*	A01:23638077-23639663	5.82	49,216.11	434	nucleus	*AT3G16130*	*AtRopGEF13*
*BrRopGEF2*	*Bra020048*	A02:4812438-4815205	5.56	565,501	494	nucleus	*AT5G19560*	*AtRopGEF10*
*BrRopGEF3*	*Bra006510*	A03:3845994-3848462	7.75	56,166.59	488	nucleus cytosol	*AT5G19560*	*AtRopGEF10*
*BrRopGEF4*	*Bra000971*	A03:14185129-14186999	4.98	53,899.85	478	nucleus	*AT4G00460*	*AtRopGEF3*
*BrRopGEF5*	*Bra013249*	A03:19623928-19627223	5.66	58,016.66	518	nucleus	*AT3G24620*	*AtRopGEF8*
*BrRopGEF6*	*Bra004945*	A05:2591812-2594623	4.95	51,723.86	460	nucleus	*AT2G45890*	*AtRopGEF4*
*BrRopGEF7*	*Bra005322*	A05:4825068-4826940	6.06	44,107.32	391	nucleus	*AT1G31650*	*AtRopGEF14*
*BrRopGEF8*	*Bra005323*	A05:4825068-4826943	6.6	56,592.05	501	nucleus	*AT1G31650*	*AtRopGEF14*
*BrRopGEF9*	*Bra030396*	A05:10855878-10857882	5.61	60,575.65	536	nucleus	*AT1G52240*	*AtRopGEF11*
*BrRopGEF10*	*Bra027189*	A05:19599734-19601858	5.68	60,452.88	530	nucleus	*AT3G16130*	*AtRopGEF13*
*BrRopGEF11*	*Bra018956*	A06:1021072-1023077	5.64	60,598.73	536	nucleus	*AT1G52240*	*AtRopGEF11*
*BrRopGEF12*	*Bra015068*	A07:3804713-3807050	5.65	58,233.8	521	nucleus	*AT3G24620*	*AtRopGEF8*
*BrRopGEF13*	*Bra015010*	A07:4301480-4303430	5.84	60,516.64	547	nucleus	*AT4G38430*	*AtRopGEF1*
*BrRopGEF14*	*Bra003536*	A07:13678585-13681092	5.5	57,786.29	513	nucleus cytosol	*AT1G79860*	*AtRopGEF12*
*BrRopGEF15*	*Bra037342*	A09:971667-973569	5.03	53,733.8	477	nucleus cytosol	*AT4G00460*	*AtRopGEF3*
*BrRopGEF16*	*Bra036671*	A09:5723788-5725881	5.86	56,589.91	505	chloroplast	*AT3G24620*	*AtRopGEF8*
*BrRopGEF17*	*Bra023198*	A09:20826415-20833784	6.41	106,720.03	941	cytosol	*AT1G31650*	*AtRopGEF14*
*BrRopGEF18*	*Bra007183*	A09:27809243-27812082	8.86	63,303.85	571	cytosol	*AT3G55660*	*AtRopGEF6*
*BrRopGEF19*	*Bra002246*	A10:10684361-10686961	6.65	59,090.48	517	chloroplast	*AT5G19560*	*AtRopGEF10*
*BrRopGEF20*	*Bra009152*	A10:15438568-15440923	9.12	66,383.94	595	nucleus	*AT5G05940*	*AtRopGEF5*
*BrRopGEF21*	*Bra009621*	A10:17469516-17471749	5.61	63,438.88	521	nucleus	*AT5G02010*	*AtRopGEF7*

**Table 2 ijms-25-03541-t002:** Information on motifs of RopGEFs in *B. rapa*.

Motif	Motif Consensus
Motif1	EMMKERFAKLLLGEDMSGGGKGVCSALALSNAITNLAASVFGEQWRLZPL
Motif2	FPGJPQSSLDISKIQYNKDVGKAILESYSRVLESLAYTILSRIEDVLYAD
Motif3	QKDSVNQVLKAAMAINAQVLSEMEIPESYJDSLPKNGKASLGDSIYKMJT
Motif4	QQTNKBGTSTEIMTTRQRSDLLMNJPALRKLDSMLJDTLDS
Motif5	EMFDPDQFLSSLDLSSEHKALDLKNRIEASIVIWKRKMVZK
Motif6	QRNDEKWWLPVVKVPPNGLSEESRKFLQS
Motif7	RWRREMDWLLSVTDHIVEFVP
Motif8	SPWGSAVSLEKRELFEERAETJLVLLKQR
Motif9	KDQTEFWYVERDSEE
Motif10	PTKSPRVTPKKLSYLEKLENMRSPTARH

## Data Availability

Data are contained within the article/Appendix A.

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
