# Peer review of "The RopGEF Gene Family and Their Potential Roles in Responses to Abiotic Stress in Brassica rapa"

_ijms, 2024, doi:10.3390/ijms25063541_

Round 1
Reviewer 1 Report
Comments and Suggestions for Authors
Dear authors!
In this peer-reviewed paper, genes of the poorly studied very important RopGEF Gene Family have been identified and analysed in the Brassica rapa. The proteins encoded by these genes play key roles in plant root growth, pollen tube growth, and responses to phytohormones and abiotic stress. Numerous currently well-developed bioinformatic approaches have been used to provide a comprehensive characterisation of the genes in this family. The analysis of the physical-chemical properties and subcellular localisation of proteins encoded by the RopGEF Gene Family was performed. Conserved domains characteristic for these proteins were revealed. The exon-intron structure of the genes was studied, and the cis-elements of the promoters of the studied genes were analysed. The localisation of the genes on Brassica rapa chromosomes was shown. Using transcriptome analysis and RT-qPCR, gene expression under salinity and osmotic stress was studied.
There are quite a lot of such articles nowadays. This article provides new information on the poorly studied RopGEF gene family. The article is useful.
I have several comments on this manuscript.
1). This paper did not study drought, per se, but PEG-induced osmotic stress, one of the negative consequences of which is water deficit. Water deficit is preferably characterised. This could be tissue water content, relative water content, and osmotic and/or water potential. Given that the article is not a detailed study of drought, I would recommend that all the places in the text that talk about drought be replaced with osmotic stress. It won't be entirely accurate, but it will still be better than drought, which the authors didn't actually create.
2) Why did the authors take two week old Chinese cabbage seedlings as a control, while the experimental plants were 12 hours older (exposure time to PEG-6000 and 150 mM NaCl). I don't know if there is a significant difference between these plants, but the set-up of the experiment is not entirely accurate (lines 483-487).
3) I cannot agree with the authors of the manuscript who make no distinction between organ and tissue of a plant. For them, any organ is just a tissue. However, everyone knows that an organ is usually composed of several or many tissues. Such confusion goes in so many places in the text. If even some computer programme makes the author call an organ a tissue, the final decision must be made by a human being. Everything needs to be corrected. (Table S2, S3 and many other places in the manuscript).
4). Table 3 says ID, but no ID values are given. Perhaps an edit needs to be entered, but it is possible that I am not fully understanding this.
Author Response
Thank you for your letter and for the reviewers' comments concerning our manuscript entitled "The RopGEF Gene Family and their Potential Roles in Responses to Abiotic Stress in Brassica rapa" (ID: ijms-2908003). Those comments are all valuable and very helpful for revising and improving our manuscript, as well as the important guiding significance to our research. We have studied comments carefully and have made the correction which we hope meet with approval. The revised portion is highlighted in yellow in the manuscript. We have submitted the revised content to the attachment, please review it.

Reviewer 2 Report
Comments and Suggestions for Authors
Dear Authors,
The paper titled "RopGEF Gene Family and Their Potential Roles in Abiotic Stress Responses in Brassica rapa" conducted a comprehensive analysis of BrRopGEF genes in B. rapa and their functions in response to abiotic stress.
Detailed Comments:
1.The abstract is well-structured.
2. The introduction concludes with the aim of the study, but it is not clearly explained.
3. The Materials and Methods section is adequately documented.
4. Interpretation of Results: The authors discuss the significance of the obtained data from various experiments, including gene expression analysis, protein structures, and protein-protein interaction networks. They attempt to understand how these results may influence the understanding of BrRopGEF functions in response to abiotic stress. The authors identify genes potentially important for plant tolerance to drought and salt and analyze their protein structures and interactions. Although the study is primarily based on bioinformatics analysis, it provides valuable insights into plant adaptation mechanisms to stressful conditions. However, experimental confirmation of the results and a more detailed analysis of protein structures could strengthen the conclusions and provide a more comprehensive understanding of BrRopGEF gene functions.
5. The discussion analysis in the paper appears to be well-conducted as the authors strive to understand and interpret the significance of their results and place them in the context of existing scientific knowledge. Here are a few elements contributing to its quality:
− Theoretical Context: The authors seek to understand the significance of their results in the context of existing scientific literature, referring to previous studies and hypotheses.
− Open Questions: The discussion also includes pointers towards future research directions, indicating questions that remain open and require further experiments for a fuller understanding.
6. Conclusions and Implications: The authors draw conclusions based on their results and discuss their implications for further research and potential practical applications, such as breeding stress-resistant plants.
Strengths of the Paper:
1.Identification and Analysis of BrRopGEF Genes: The paper conducts a comprehensive identification of 21 BrRopGEF genes in the B. rapa genome and analyzes their physicochemical properties and subcellular localization.
2. Analysis of Protein Structures: The paper utilizes bioinformatics tools to predict BrRopGEF protein structures, allowing for a better understanding of their functions.
3. Gene Expression Analysis in Response to Abiotic Stress: The study utilizes RNA-seq data and RT-qPCR analysis to examine BrRopGEF gene expression in response to drought and salt stress, helping to identify genes significant for plant adaptation to stressful conditions.
4.Overall, the paper provides valuable insights into the roles of BrRopGEF genes in abiotic stress responses, supported by comprehensive analyses and interpretations. However, further experimental validation and detailed protein structure analysis could enhance the robustness of the findings.
Weaknesses of the study:
1.Lack of experimental validation: Despite extensive bioinformatic analysis, there is a lack of experimental validation of the functions and interactions of BrRopGEF genes. In vitro or in vivo studies could enhance the credibility of the results.
2. Limited analysis of protein structures: The study primarily focuses on predicting the protein structures of BrRopGEF, but lacks a more detailed analysis of functional protein domains.
Here are the suggestions:
1. Expanded analysis of protein structures: It is advisable to conduct a more detailed examination of functional protein domains. This can be achieved through a more thorough structural and functional analysis of the individual protein domains within the studied BrRopGEF genes.
Comments on the Quality of English LanguageMinor editing of English language required
Author Response

(The authors gave the same response as above.)
